# Solid waste management practices and challenges in Besisahar municipality, Nepal

**Mahendra Aryal** [1]*, **Sanju Adhikary**[2]

**1** Department of Chemistry, Tribhuvan University, Tri-Chandra Campus, Kathmandu, Nepal, **2** Department of Economics, School of Law, Kathmandu University, Dhulikhel, Nepal

\* mahendraaryalnp@yahoo.com

**Data Availability Statement:** The manuscript encompasses all relevant data, along with its accompanying Supporting Information files.

**Funding:** The author(s) received no specific funding for this work.

## Abstract

This study is a comprehensive assessment of the waste management system in Besisahar municipality. Information and some data have been collected from the municipality of Besisahar, followed by interviews with municipal officials responsible for waste management, stakeholders, waste workers, and residents. A total of 230 households, 20 schools, 10 government and private offices, 10 financial institutions, 60 commercial hotels, restaurants, and shops, and 20 medical shops and healthcare institutions, were selected in this study by random sampling. An extensive field study was conducted within all municipal wards and at dump sites. The results indicated that 42.14% of solid waste was collected through door-to-door collection services, 5.87% was mismanaged in open public places, 11.21% was used as compost manure, and the rest was discarded on riverbanks, dug up, and burned. A large component of the characterization of household waste consisted of organic waste (68.03%), followed by paper/paper products (8.13%), agricultural waste (5.5%), plastic (5.21%), construction (3.81%), textile (2.72%), metals (0.54%), glass (1.01%), rubber (0.10%), electronic (0.05%), pharmaceutical (0.1%) and others (4.78%) in the Besishahar municipality. Solid waste generation was found to be at 197.604 g/capita/day, as revealed by cluster sampling in 230 households. Around 4.285 tons-solid waste/day were generated in urban areas, while 16.13 tons-solid waste/day was estimated for the whole municipality. An important correlation between the parameters of solid waste was found by statistical analysis. Currently, solid waste is dumped on riverbanks, open fields, and springs, creating environmental and health hazards. The findings of this study will be useful to Besisahar municipality and its stakeholders in forming policies that facilitate waste management practices in this region and promote sustainable waste management systems.

## Introduction

Minimizing solid waste management (SWM) has become a global challenge due to limited resources, an increasing population, and rapid urbanization and industrialization [1]. In developing countries, most waste is composed of organic materials, which are generally three times higher than in industrialized countries [2]. The improper disposal of electronic waste,

**Competing interests:** The authors have declared that no competing interests exist.

polythene materials, and medical waste in the locality also contributes to pollution and public health hazards [3–5]. Waste management (WM) involves several interdependent activities such as collection, source separation, storage, recycling and reuse, biological treatment, transportation etc. that are coordinated and controlled efficiently [6]. Moreover, waste recycling reduces greenhouse gas emissions and particulates from incinerators, and increases the lifespan of landfill [7, 8].

In Nepal, the Solid Waste Management Act, 2011 AD, and the Solid Waste Management Rules, 2013 AD, serve as the governing regulations for the SWM sector. These regulations establish the guidelines and framework for managing solid waste throughout the country. The primary purpose of these regulations is to provide guidelines and structure for the proper management of solid waste, aiming to maintain a clean and healthy environment while minimizing its adverse impacts on public health and the surroundings [9, 10]. According to these regulations, local governments are entrusted with the responsibility of collecting, transporting, and disposing of waste in an environmentally sustainable manner. Their role includes promoting recycling, reusing, and reducing municipal solid waste (MSW) at the source, along with the necessary segregation of waste [11, 12]. Also, the Act permits the imposition and collection of service fees against solid WM services and specifies principles for fixing such charges and procedures for collecting and utilizing them [9]. Solid waste collection, treatment, and final disposal are all the responsibility of Nepalese municipalities [1]. Moreover, competitive bidding allows non-public organizations, community-based organizations, and non-government organizations to be involved in SWM [9].

Sustainable Development Goals (SDGs) were established as a result of the UN Conference on Sustainable Development in June 2012 and the UN General Assembly in September 2014. In the SDGs, 17 goals were set up for reducing poverty, enhancing social equality, decreasing pollution levels, and making cities more livable [13]. To achieve sustainability, the Global Waste Management Goals include ensuring affordable, secure, and accessible solid waste collection services; preventing open burning and dumping; and managing environmentally sound WM by 2030 [14, 15]. Nepal is committed to the current global initiative as a member of the UN [16].

This study focuses on examining the unique context of SWM in Besisahar municipality. It investigates WM techniques, infrastructure, resources, policies, and challenges specific to the area. The study evaluates present WM systems, including collection, transportation, and disposal. It identifies challenges such as inadequate garbage collection systems, a lack of public awareness, and financial limitations. In addition, it also provides valuable insights for policymakers, WM practitioners, and researchers by addressing these challenges and proposing recommendations and solutions to improve SWM in Besisahar. The study aims to promote sustainable and efficient WM strategies for the benefit of the municipality's residents and the environment.

## Methodology

### Ethics statement

This study was conducted with permission from the municipality of Besisahar. Data collection began with a brief explanation of the purpose of the study and the right to participate or decline. It ensured confidentiality, privacy, and anonymity for the participants. Each participant provided written and verbal consent prior to data collection.

### Study area

Besisahar is the headquarter of Lamjung district, and the study area covers the village and urban areas with dense populations. There are 11 political wards in BM, covering 127.64 km$^2$

**Table 1. Total population of BM [17, 18].**

| Ward no. | Name of ward | No. of family | Male | Female | Total | % |
|---|---|---|---|---|---|---|
| 1 | Udipur | 570 | 1424 | 1353 | 2777 | 6.23 |
| 2 | Bhakunde | 906 | 2146 | 2075 | 4221 | 9.46 |
| 23 | Gaunsahar | 756 | 1739 | 1672 | 3411 | 7.65 |
| 4 | Purankot | 247 | 693 | 682 | 1375 | 3.08 |
| 5 | Nalma | 382 | 974 | 856 | 1830 | 4.10 |
| 6 | Chandisthan | 587 | 1416 | 1337 | 2753 | 6.17 |
| 7 | Besishahar | 1994 | 4320 | 4383 | 8703 | 19.51 |
| 8 | Besishahar | 1881 | 4008 | 3864 | 7872 | 17.65 |
| 9 | Besishahar | 497 | 1176 | 1150 | 2326 | 5.22 |
| 10 | Banjhakhet | 795 | 1762 | 1585 | 3347 | 7.50 |
| 11 | Chiti | 1219 | 3073 | 2910 | 5983 | 13.42 |
| Total | | 9834 | 22731 | 21867 | 44598 | 100 |

of geographical area. It lies between latitude 28˚ 23' 26.41" N and longitude 84˚ 34' 28.13" E. The elevation ranges from 728 to 1268 meters above sea level, with a mean annual temperature of 25˚C.

## Population of BM

Besisahar municipality consists of high-, medium-, and low-density residential areas. Several government offices, schools, colleges, hospitals, industries, hotels, agricultural farms, non-governmental organizations, and religious and cultural landmarks are located in the BM. A total of 44,598 people lived in BM, including 22731 males and 21867 females [17, 18]. Out of the total wards, the highest number of residents was 8703 in ward number 7, while the fewest number of residents was 1375 in ward number 4. There were 9834 households in BM. Ward 8 had the highest number of households with a total of 1994, and Ward 4 had the lowest with just 247 households (Table 1). According to the BMP [17], the majority of residents in BM were between the ages of 10 and 14, whereas the least populated age group was between 70 and 74.

## Research method

To conduct the present study, both primary and secondary sources of information including residential, commercial, institutional, industrial, agricultural, construction and demolition areas activities, published reports of BM, National Population and Housing Census (NPHC), Asian Development Bank (ADB), National Economic Census (NEC), and other verifiable and credible internet sources were used. Observations, interviews, and questionnaire surveys were conducted during January–May 2022 AD. Primary and secondary data were obtained through interviews with various officials of the BM of all political wards, WM employees, residents of the study areas, academic and commercial establishments, as well as relevant stakeholders (See S1 File).

A cluster sampling technique was used to collect data from 230 households. The questionnaires were prepared about types of waste and their composition, collection, and management. Participants were asked to describe their socioeconomic status, the types of waste generated by their households, resource management, as well as how they stored and disposed of their solid waste. About 52.17% of the sample size (120) represented rural households located in wards 4, 5, 6, 9, 10, and 11, 26.08% of the sample size (60) represented semi-urban households located in wards 1, 2, and 3, while 21.74% of the sample size (50) represented urban households located

in wards 7, and 8, respectively. Each cluster consisted of ten households. Twenty-three clusters were examined in total. As part of the study, 20 schools, 10 government and private offices, 10 financial institutions, 60 commercial hotels, restaurants, and shops, as well as 10 medical shops and 10 healthcare institutions, were selected through a random sampling method. There were separate questionnaires prepared for the survey of households and the survey of the institutional and commercial sectors, as well as for the municipality office to collect data related to SWM.

## Determination of waste

To educate households, hotels, restaurants, shops, government and non-government offices, schools, and hospitals about the importance of collecting waste within 24 hours, pre-visits were conducted at selected locations and the waste collection bags were provided to them. Solid wastes were then segregated into organic wastes, plastics, paper products, metals, glass, rubber, textiles, construction debris, hazardous wastes, and other wastes. The amounts of different wastes collected from homes, hotels, hospitals, schools, factories, offices, and governmental and private organizations were measured on a wet weight basis with a digital balance.

## Data analysis

General data were analyzed using descriptive statistics. The analysis of the data was carried out using both quantitative (waste generation rates, cost-benefit analysis, and statistical analysis) and qualitative (interviews, case studies, and field surveys) methods. In this study, a significance level of $p < 0.05$ was adopted as the threshold for determining statistical significance. Arithmetic mean, median, average deviation, standard deviation, skewness and kurtosis were determined. A Pearson correlation analysis (r) was conducted to determine the strength of the relationship between parameters related to SWM. Interviews and observation frames yielded qualitative information that was coded, classified, and analyzed into different themes. Microsoft Excel was used to analyze the data.

# Results and discussion

## Waste generation and segregation

It is necessary to handle, store, collect and dispose of waste generated by human activities to reduce the risk of environmental and public health problems [19]. Waste segregation at the source is considered an important practice in solid waste management. However, studies have indicated low levels of waste segregation among households in Nepal [20]. The composition of solid waste generated in both rural and urban areas of BM was categorized and presented in Table 2. The most prevalent waste types were found to be food, paper, plastic, rubber, wood, metals, glass, hazardous, biomedical, agricultural, and industrial wastes; however, due to socioeconomic factors and cultural practices, the types and rates of generation of waste vary greatly from place to place [1, 19, 20]. As seen in Fig 1, different types of waste were transported to the nearby collection centre for urban areas. Some residents actively separate certain types of waste such as paper and metals due to the direct economic benefits. The findings of the questionnaire study indicated that a considerable portion of households, approximately 72%, possessed knowledge about waste separation. However, despite being aware of this concept, they did not translate their knowledge into practical implementation. Conversely, only 28% of the respondents indicated actively applying their knowledge to properly separate waste at the household level.

**Table 2. Sources of solid wastes in BM.**

| Type of Waste | Source/Waste generator |
|---|---|
| Agricultural wastes, animal waste, electronic waste, paper, plastic, glass, and metals. | Residential |
| Plastic bags, manure, oil, fertilizer, pesticides, containers, chemicals, and plastics. | Agricultural |
| Plastic, glass, cans, paper, plastic, construction waste, electronic waste, and food waste. | Commercial |
| Construction waste, food waste, glass, tin cans, paper, plastics, and electronics waste. | Institutional |
| Biological, surgical, pharmaceutical, food, hazardous waste, paper, plastics, metal cans, glass, construction and electronic waste. | Hospital |
| Demolition and construction waste, hazardous waste, glass, plastic, paper, tin cans, metal, ash, rubber, containers, chemicals, and digital waste. | Industrial |

Several waste types have been studied in some studies, encompassing organic waste, plastics, paper and paper products, metals, glass, rubber and leather, textiles, dirt and construction debris, hazardous wastes, and other forms of waste [9, 21]. Notably, the classification of municipal solid wastes was expanded by Dangi et al. [11] and Dangi et al. [22] to include two new waste types, namely hazardous wastes, and dirt and building debris.

By categorizing their solid waste into several categories for collection and treatment, and encouraging the 3Rs (reduce, reuse, recycle), households can significantly improve waste management. A three-bin system, which allows residents to divide waste into dry recyclables, moist organic and/or domestic waste, and hazardous or residual waste, has been implemented in several places [9]. In Nepal, initiatives to encourage trash segregation have been gaining momentum. However, insufficient infrastructure, a lack of understanding, and restricted access to waste management services are frequently blamed for poor segregation practices. To improve waste segregation practices, it is essential to raise public awareness of its advantages, educate

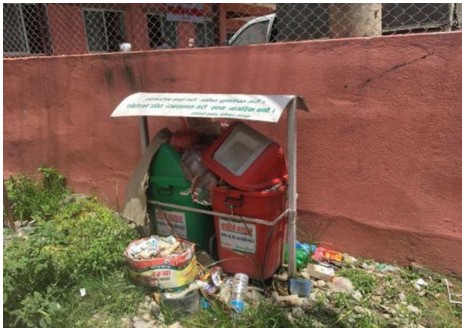
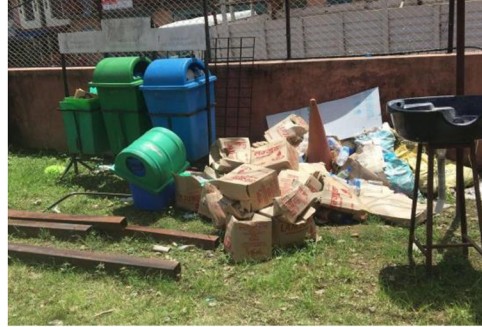
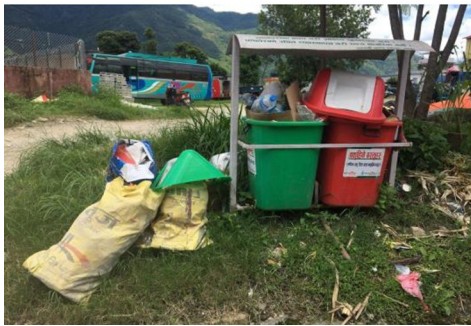
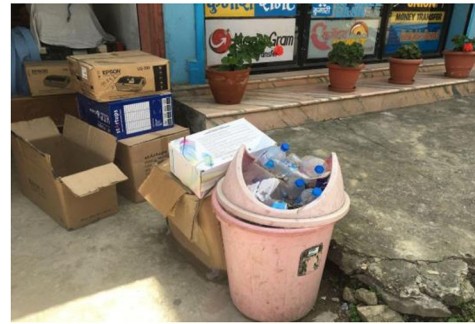

**Fig 1. Collection of solid waste in urban areas.**

people about it, encourage public engagement, build segregation facilities, and develop a waste management system.

## Household SWM practices

Open dumping in public places or along the banks of rivers or streams is a traditional method of handling solid waste in the BM. An overview of household SWM in BM is provided in Table 3. Many village households had their dumping sites where they made compost and only a small amount of waste was disposed of outside. It was observed that some households in villages disposed of waste near public places. A large number of households did not separate their waste when dumping it. Households even threw garbage into urban areas without respecting the collection centers. Several households and individuals were observed to be less aware of where to dump their waste. The field survey indicated that solid waste collection had been effectively implemented in wards 1, 2, 3, 7, 8, and 9, respectively. As of now, there is no waste collection service in wards 4, 5, 6, 10, and 11. In addition, wards no. 7, 8, and 9 of urban areas had a door-to-door solid waste collection service, which was found to be at 87.8, 94.4, and 10.3%. Ward no. 5 had the lowest rate of digging and burning solid waste and residents managed in their compounds (3.4%), while ward no. 8 had the highest rate (80.2%). The proportion of waste managed by households and sent to waste collection centers varied from 3.4% in ward no. 7 to 60.6% in ward no. 10. The highest rate of waste dumped in nearby spring-fed rivers or channel runoff was found at 64.4% in ward no. 5, and the lowest was at 0.2% in ward no.

**Table 3. Household SWM in BM [17].**

| Ward no. | Family / % | SWM | | | | | | |
|---|---|---|---|---|---|---|---|---|
| | | Door-to-door service | Digging and burning in own garden | Transfer to a waste collection centre | Dumping to nearby spring-fed rivers or channel runoff | Use for compost manure | Public place | Others |
| 1 | Household | 160 | 329 | 49 | 92 | 221 | 9 | 0 |
| | % | 28.1 | 57.7 | 8.6 | 16.1 | 38.8 | 1.6 | 0 |
| 2 | Household | 297 | 457 | 137 | 90 | 887 | 31 | 2 |
| | % | 32.8 | 50.4 | 15.1 | 9.9 | 49.3 | 3.4 | 0.2 |
| 3 | Household | 73 | 514 | 241 | 129 | 86 | 23 | 0 |
| | % | 9.7 | 68 | 33.3 | 17.1 | 11.4 | 3 | 0 |
| 4 | Household | 0 | 41 | 62 | 152 | 1 | 1 | 0 |
| | % | 0 | 16.6 | 25.1 | 61.5 | 0.4 | 0.4 | 0 |
| 5 | Household | 4 | 13 | 121 | 246 | 3 | 107 | 0 |
| | % | 1.0 | 3.4 | 31.7 | 64.4 | 0.8 | 28.0 | 0 |
| 6 | Household | 32 | 311 | 323 | 29 | 5 | 17 | 0 |
| | % | 5.5 | 53.0 | 55.0 | 4.9 | 0.9 | 2.9 | 0 |
| 7 | Household | 1750 | 172 | 67 | 44 | 92 | 0 | 5 |
| | % | 87.8 | 8.6 | 3.4 | 2.2 | 4.6 | 0 | 0.3 |
| 8 | Household | 1775 | 69 | 409 | 4 | 9 | 4 | 0 |
| | % | 94.4 | 3.7 | 21.7 | 0.2 | 0.5 | 0.2 | 0 |
| 9 | Household | 51 | 289 | 158 | 128 | 52 | 187 | 0 |
| | % | 10.3 | 58.1 | 31.8 | 25.8 | 10.5 | 37.6 | 0 |
| 10 | Household | 1 | 281 | 482 | 202 | 6 | 46 | 1 |
| | % | 0.1 | 35.3 | 60.6 | 25.4 | 0.8 | 5.8 | 0.1 |
| 11 | Household | 1 | 978 | 469 | 148 | 180 | 152 | 1 |
| | % | 0.1 | 80.2 | 38.5 | 12.1 | 14.8 | 12.5 | 0 |

Note: Percentage is greater than 100 due to the possible multiple answers.

**Table 4. Households SWM with the total family number and percentage in BM [17].**

| WM | Family no. | Percentage |
|---|---|---|
| Door-to-door service | 4144 | 42.14 |
| Digging and burning in own garden | 3454 | 35.12 |
| Waste collection centre | 2528 | 25.71 |
| Nearby spring-fed rivers or channel runoff | 1264 | 12.85 |
| Use for compost manure | 1102 | 11.21 |
| Public places | 577 | 5.87 |
| Others | 9 | 0.09 |
| Total | 13078 | 132.99 |

Note: Percentage is greater than 100 due to the possible multiple answers.

8. For the use of compost manure, a minimum and maximum range was determined at 0.4 and 49.3% in wards no. 4 and 2, while for mismanagement in public places, 0 and 37.6% were determined in wards 7 and 9, respectively.

Table 4 shows household SWM with total family number and percentage in BM. Among households, only 42.14% had door-to-door household waste collection services, while 5.87% had mismanaged their solid waste in public places. In addition, only 11.21% of solid wastes were used as compost manure, with most of the rest being discarded on riverbanks, in public places, dug up, and burned. Household waste was seen being dumped into water bodies directly, which can harm the environment and the general public's health. The overall results indicated that the WM system of the BM was ineffective and unscientific. WM practices should be implemented in such a way that the result of waste in the external environment is minimized [11]. The segregation of these wastes by waste generators is a necessary component of a successful SWM system, but it is not done in the BM.

## Households waste characterization

Most of the food waste generated in villages was used as animal feed, and organic waste was relatively low in comparison to urban areas. The waste generation, including paper, plastic, rubber, electronics, pharmaceuticals, etc., was much higher in urban areas compared to villages. During the field survey, electronic and hazardous waste were observed in public places, and it showed that people in the urban and village areas have less awareness regarding the categories of waste settlement. It was noticed that neither the municipality nor the private waste collectors characterize waste, and neither has a system to determine how much waste has been generated and collected.

An assessment of the household waste composition generated in each ward of the Besisahar municipality is given in Table 5. The results showed that almost all of the waste generated by households was organic (68.03%). The results further revealed that besides organic wastes, 5.5% of the wastes were agricultural wastes, 3.81% construction debris, 5.21% plastics, 1.01% glass, 8.13% paper, 2.72% textiles, 0.54% metals, 0.1% rubber, 0.1% pharmaceuticals, 0.05% electronics, and 4.78% other wastes.

The waste management study draws attention to the significant diversity in the Median, AVEDEV, and STDEV values. The range of median values, from 0.23 to 372, highlights the variety of results across waste management strategies. There are various levels of deviation from the mean in waste management practices in BM, as indicated by the AVEDEV range of 0.293 to 196.82. The STDEV values range from 0.375 to 250.9, revealing a wide range of practices with various degrees of consistency.

**Table 5. Assessment of the household waste composition generated in each ward of the Besisahar municipality.**

| Waste type | Minimum and maximum waste generation in BM's wards | | Mean (Kg/day) | Total waste in BM (Kg/day) | % | Statistical parameters | | | | |
|---|---|---|---|---|---|---|---|---|---|---|
| | Min (Kg/day) | Max (Kg/day) | | | | Median | AVEDEV | STDEV | Skewness | Kurtosis |
| Organic | 125 (4)* | 912 (7)* | 449.82 | 4948.02 | 68.03 | 372 | 196.82 | 250.9 | 0.788 | -0.196 |
| Paper | 16.25 (5)* | 139 (8)* | 58.04 | 638.5 | 8.13 | 45.1 | 29.85 | 39.42 | 0.941 | 0.707 |
| Agricultural | 27.35 (3)* | 62.89 (11)* | 38.31 | 421.44 | 5.5 | 34.07 | 7.62 | 10.44 | 1.32 | 2.23 |
| Plastics | 9.3 (4)* | 80 (8)* | 9.64 | 412.38 | 5.21 | 37.48 | 17.55 | 23.63 | 0.477 | -0.676 |
| Construction | 7.13 (5)* | 72.14 (8)* | 26.72 | 293.95 | 3.81 | 17.1 | 18.54 | 22.87 | 1.11 | 0.201 |
| Textile | 7.5 (5)* | 41 (7)* | 19.5 | 214.5 | 2.72 | 12 | 11.72 | 13.47 | 0.736 | -1.23 |
| Metals | 1.26 (5)* | 8.51 (7)* | 3.89 | 42.84 | 0.54 | 3.79 | 31.71 | 2.24 | 0.586 | 0.255 |
| Glass | 3.32 (3)* | 19 (7)* | 7.25 | 79.84 | 1.01 | 5.1 | 3.90 | 5.04 | 1.32 | 1.81 |
| Rubber | 0.13 (4)* | 3.01 (7)* | 0.782 | 8.61 | 0.109 | 0.26 | 0.782 | 1.02 | 1.45 | 1.57 |
| Electronic | 0102 (5)* | 1.12 (8)* | 0.391 | 4.30 | 0.055 | 0.23 | 0.293 | 0.375 | 1.25 | 0.728 |
| Pharmaceutical | 0.12 (3)* | 3.08 (8)* | 0.761 | 8.374 | 0.108 | 0.35 | 0.67 | 0.937 | 1.64 | 3.28 |
| Others | 14 (4)* | 64 (7)* | 32.27 | 355 | 4.78 | 28 | 13.07 | 16.59 | 0.817 | -0.195 |

Note: *: Total waste generated in respective ward numbers.

In general, a normal distribution of waste generation has coefficients of skewness and kurtosis of 0 and 3, respectively [11, 23, 24]. An asymmetric distribution is measured by the skewness coefficient. Perfect symmetry is indicated by a coefficient of zero, while positive and negative skewness suggests longer right and left tails. Our values for skewness range from 0.477 to 1.64, the distribution is moderately skewed and is approximately symmetric. It shows that all waste types had a positive skewness (to the right). The kurtosis coefficient measures the peakedness or flatness of a distribution. The calculated values of kurtosis range from -0.195 to 3.28, where positive kurtosis indicates heavier tails and the presence of outliers, while negative kurtosis suggests lighter tails and fewer outliers. Overall values of both coefficients showed the normal distribution of waste generation in BM.

Studies have consistently shown that a significant proportion of municipal solid waste is organic in nature. The composition of household waste in Jeetpur Simara Sub-Metropolitan City was estimated to be 80% organic, 10% plastic, 6% paper, and 4% inert [25]. Among 53 municipalities in Nepal, organic waste had the highest household waste composition at 66%, followed by plastics at 12%, and paper products and paper waste at 9% [9]. Pokhrel and Viraraghavan [1] found that more than 70% of the waste fraction falls into this category. ADB [9] reported that organic waste accounts for 56% of the total waste generated, excluding agricultural waste. In the specific context of Pokhara Metropolitan City, it was found that organic waste makes up the largest fraction, constituting 73% of the total waste generated, followed by inorganic waste at 24%, and other wastes at 3%. Similar patterns were observed in Janakpurdham Sub-Metropolitan City, where organic waste comprised the largest fraction, followed by inorganic waste and other wastes [20]. Even though the high level of food percentage in Nepal, is still lower than in Iran (Rasht city) with a value of 85% [17]. Our findings align with the waste composition results reported by Das et al. [26] for Pokhara Metropolitan City.

### Pearson correlation matrix between population and various wastes

The Pearson correlation matrix is a useful tool for analyzing relationships in the field of solid waste management. Researchers and waste management professionals can build efficient waste management strategies by understanding the relationships between population and

**Table 6. Pearson's correlation matrix for population and SW in different wards of BM (r: range = -1.00 to +1.00).**

| | Population | Organic | Paper | Plastics | Metals | Textile | Glass | Rubber | Agricultural | Construction | Pharmaceutical | Electronic | Others |
|---|---|---|---|---|---|---|---|---|---|---|---|---|---|
| Population | 1 | | | | | | | | | | | | |
| Organic | 0.798 | 1 | | | | | | | | | | | |
| Paper | 0.851 | 0.967 | 1 | | | | | | | | | | |
| Plastics | 0.793 | 0.927 | 0.911 | 1 | | | | | | | | | |
| Metals | 0.779 | 0.855 | 0.861 | 0.973 | 1 | | | | | | | | |
| Textile | 0.743 | 0.944 | 0.924 | 0.926 | 0.84 | 1 | | | | | | | |
| Glass | 0.81 | 0.952 | 0.977 | 0.91 | 0.873 | 0.938 | 1 | | | | | | |
| Rubber | 0.845 | 0.942 | 0.942 | 0.884 | 0.851 | 0.897 | 0.964 | 1 | | | | | |
| Agricultural | 0.141 | -0.048 | 0.038 | -0.183 | -0.26 | 0.004 | -0.023 | -0.151 | 1 | | | | |
| Construction | 0.77 | 0.965 | 0.925 | 0.922 | 0.852 | 0.939 | 0.921 | 0.954 | -0.219 | 1 | | | |
| Pharmaceutical | 0.818 | 0.884 | 0.856 | 0.819 | 0.736 | 0.86 | 0.835 | 0.918 | -0.161 | 0.946 | 1 | | |
| Electronic | 0.835 | 0.951 | 0.933 | 0.926 | 0.878 | 0.926 | 0.941 | 0.982 | -0.21 | 0.986 | 0.952 | 1 | |
| Others | 0.826 | 0.987 | 0.987 | 0.94 | 0.875 | 0.956 | 0.97 | 0.944 | -0.02 | 0.956 | 0.889 | 0.953 | 1 |

other waste parameters. Understanding these relationships enhances decision-making and improves waste management approaches.

The researchers have employed a Pearson's correlation matrix to explore the degree of association among various factors in waste management practices. The correlation values, ranging from 0.7 to 1, indicate a strong correlation, while values between 0.5 and 0.7 suggest a moderate correlation [27]. Table 6 presents the Pearson correlation coefficients for different SWM parameters in each ward of BM. The analysis revealed significant associations within the context of SWM.

Population was found to be strongly correlated with several waste types, including organic (r = 0.798), paper (r = 0.851), plastic (r = 0.793), metal (r = 0.779), textile (r = 0.743), glass (r = 0.810), rubber (r = 0.845), construction materials (r = 0.770), pharmaceutical (r = 0.818), electronic (r = 0.835), and other wastes (r = 0.826). These results indicate that generating organic, paper, plastic, metal, textile, glass, rubber, and construction waste consistently increases with population, highlighting strong positive associations across all parameters. These relationships offer valuable perspectives on influencing factors, enabling waste management practitioners and policymakers to make well-informed decisions.

On the other hand, agricultural waste exhibited weak correlations (r = 0.141, 0.038, and 0.004) with population, paper, and textile, respectively. These specific values point to the absence of a substantial correlation and suggest that different variables may be influencing different parameters. Moreover, agricultural waste displayed a negative correlation with organic (r = -0.048), plastic (r = -0.183), metal (r = -0.260), glass (r = -0.023), rubber (r = -0.151), construction materials (r = -0.219), pharmaceutical (r = -0.161), electronic (r = -0.210), and other wastes (r = -0.020). Notably, an increase in population was found to be associated with a decrease in agricultural waste. This negative correlation between certain waste types and agricultural waste suggests a trade-off relationship, where an increase in one variable corresponds to a decrease in the other to a certain degree.

## Household waste generation

To determine the number of grams/capita/day of waste from each household, the total weight of waste from 230 household samples was divided by the number of people living in each household. It was reported that the average family size for BM was 3.78 members [17]. By comparison, Dangi et al. [11] found a family size of 5.2 in Tulsipur, Dang, and Dangi et al. [22]

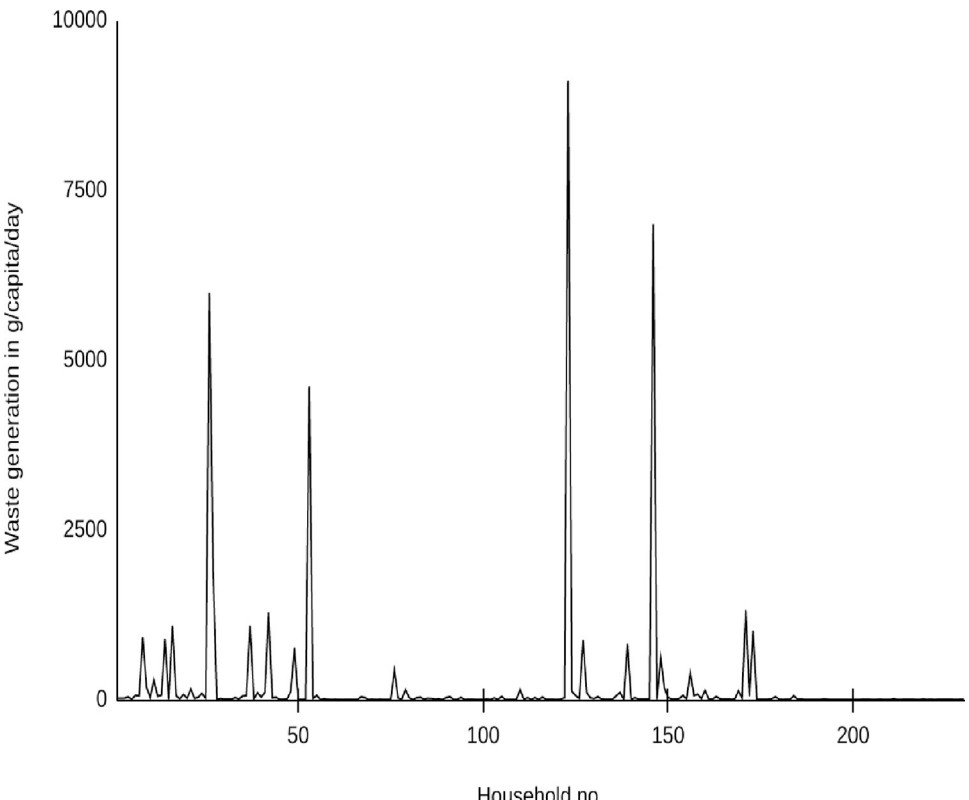

**Fig 2. Average per capita household waste generation.**

found a family size of 5.7 in Kathmandu city. According to an ADB [9] study conducted in 53 different municipalities in Nepal, the average number of family members in each family ranged between 5 and 6. In our study, the value of family size is lower than the values reported elsewhere [9, 11, 22].

The solid waste generation rate in different wards of BM is given in Fig 2. There was a significant amount of waste generated from urban households, including glass bottles, plastic, paper, rubber, metal, medicines, wood, electronic, tin, cans, construction waste, packaging materials, and hazardous waste. Compared to urban households, rural households generated mainly plastics, agricultural wastes, papers, fertilizers, some metals, and pesticides, which can be attributed to the low waste generation. The minimum and maximum average waste generation of 1.64 and 9110 g/capita/day were observed from 230 household samples. The average amount of household waste generated was calculated to be 197.604 g/capita/day for the BM. Based on the total of 3300 households from 60 municipalities with 55 households each, an average per capita household waste generation of 115 g/day was estimated by Pathak et al. [21]. The household survey in Jeetpur Simara Sub-Metropolitan City found an average household waste generation rate of 120 g/capita/day [25]. For the Tulsipur municipality, Dangi et al. [11] determined an average of 330 g/capita/day. According to ADB [9], the household waste generation rate in Nepal's 53 municipalities ranged from 75 g/capita/day to 278 g/capita/day in Triyuga and Inaruwa municipalities. The average household waste generation rate in the Triyuga municipality of Nepal was 317 g/capita/day. Possible explanations for the slightly different values include the different sample sizes taken in rural and urban areas.

**Table 7. Comparison of statistical parameters for household waste generation (g/capita/day) in Nepal.**

| Study area | Statistical parameter | | | | | | | | | Reference |
|---|---|---|---|---|---|---|---|---|---|---|
| | Sample size | Min (g/capita/day) | Max (g/capita/day) | Average (g/capita/day) | Median | AVEDEV | STDEV | Skewness | Kurtosis | |
| Besisahar Municipality | 230 | 1.64 | 9110 | 197.60 | 10.04 | 318.84 | 923.68 | 7.29 | 58.35 | Present study |
| Kathmandu Metropolitan City | 333 | - | - | 497.3 | - | - | 5349.4 | 18.2 | 332.1 | Dangi et al., 2011 |
| Tulsipur Municipality | 84 | 2.8 | 12033.3 | 330.4 | - | - | 1317.6 | 8.7 | 77.5 | Dangi et al., 2013 |

A total of 34,857 people (78.16%) were permanently residing in BM. Furthermore, 5506 people (12.35%) of the BM lived in other parts of Nepal, while 4235 people (9.50%) were abroad. It was reported that 3708 households (35.58%) had at least one member living outside the family [17, 18]. The amount of household solid waste produced in BM was 7.427 tons/day, which was obtained by multiplying the average per capita waste generation rate by the total population minus absentees in the municipality according to the 2021 census. The total household solid waste generation was calculated at 14.2 tons/day in Tulsipur municipality in Dang, Nepal [11]. In Waling municipality, there were 5,952 households and an estimated population of 51,234 individuals. The average per capita household waste generation rate in Waling was 0.68 kg/person/day, leading to a total daily household waste generation of approximately 5.51 tons. It is noteworthy that household waste comprises a significant portion, accounting for approximately 50% to 75% of the overall MSW [9].

Statistics for solid waste analysis calculated from 230 households are shown in Table 7. The median value of 10.04 represents the central value of a waste-related variable. The high AVEDEV of 318.84 suggests significant data scattering, indicating variability in waste management practices. Moreover, the STDEV of 923.68, exceeding the average deviation, indicates wide variability in waste management data, potentially due to diverse practices. The calculated coefficients of skewness and kurtosis were found to be 7.290 and 58.350, respectively. As a result of a positive skewness and a slightly higher kurtosis in this study, the distribution of waste in households appears to be random. Our results of STDEV, skewness and kurtosis coefficients are close to the values reported by Dangi et al., [11]. The results showed that most households had a lower generation rate, while a minority of households exhibited a higher mean generation rate. It was also observed that larger families resulted in a higher household waste generation rate with a lower per capita waste generation rate [11, 21].

## Degree of urbanization

The degree of urbanization may have a significant effect on per capita waste generation. As shown in Table 8, rural, semi-urban, and urban areas generate different amounts of waste. It is noted that household waste generation rates vary according to urbanization. The study found that urban households generated the highest amount of waste at 450.018 g/household/day, followed by semi-urban households at 345.44 g/household/day, and rural households at 35.771 g/

**Table 8. A comparison of the levels of urbanization and the amount of waste generated per capita.**

| Degree of urbanization | Average waste generation (g/capita/day) |
|---|---|
| Rural | 35.771 |
| Semi-urban | 345.444 |
| Urban | 450.018 |

**Table 9. Average monthly household income and expenditure in BM [17, 26].**

| Average monthly income and expenditure in US$ | Household no. | Income (%) | Household no. | Expenditure (%) |
|---|---|---|---|---|
| 0–38 | 741 | 7.54 | 692 | 7.04 |
| 38–114 | 2153 | 21.89 | 3902 | 39.68 |
| 114–190 | 1900 | 19.32 | 2826 | 28.74 |
| 190–265 | 1735 | 17.64 | 1298 | 13.2 |
| 265–379 | 1779 | 18.09 | 782 | 7.95 |
| 379–757 | 1072 | 10.9 | 259 | 2.63 |
| >767 | 454 | 4.62 | 75 | 0.76 |
| Total | 9834 | 100 | 9834 | 100 |

household/day. Accordingly, the larger the population, the higher the amount of household waste generated per day. Han et al. [28] reported that the degree of urbanization was positively correlated with MSW generation in China.

## Correlation between households' economy and waste generation

There were 1,772 establishments involved in various economic activities in BM, as reported in the 2018 Economic Census. The establishments employed 12,466 individuals, of which 6,572 were males, and 5,894 were females, as either self-employed or employees. An average of 7.03 people were employed in each business, with males accounting for 3.71 and females accounting for 3.33, respectively. The ratio of males to females engaged in economic activities in establishments was 1.12 [29].

Most households with the highest incomes reside in the urban areas of wards 7 and 8. Most households in wards 1,2, 3, 4, 5, 6, 9, 10 and 11 were in villages and had low incomes. In addition, some households of wards 1, 2, 3, and 9 were semi-urban and they had middle incomes. There were 741 households with a very low income (7.54%) a total of 9834 households. The majority of 41.21% of households were lower-middle-income, whereas 35.73% of the households were middle-income. Higher-income households were represented by 10.9%, while the highest-income household made up 4.62%. In this study, income is positively correlated to waste generation rate.

Table 9 presents the average household income/expenditures in BM. There were differences in household waste generation rates based on incomes and expenditures. Household expenditure capacity had a significant impact on waste generation. In Fig 3, it is shown that households with higher average expenditures generated more waste per day. A household with a monthly expenditure of at least US$ 767 generated approximately 950 g/household/day, whereas a household with a monthly expenditure of less than US$ 38 generated an average of 216 g/household/day. The results of this study agree with those of ADB [9] and Pathak et al. [21] in many municipalities in Nepal.

## Generation of institutional, commercial, and industrial waste

As part of the study, the average composition of institutional, commercial, and industrial waste was obtained from the analysis of waste samples. There was an average waste generation rate of 20 kg/day for industries, 7.81 kg/day for hospitals, 3.27 kg/day for hotels and restaurants, 2.5 kg/day for commercial shops, 1.67 kg/day for academic institutions, 0.43 kg/day for banks and financial institutions, and 0.32 kg/day for government and private offices. The waste composition analysis in all the establishments studied revealed 11.02% organic waste, 34% paper and paper products, 18% plastics, 8.32% textiles, 6.6% construction debris, 0.8%

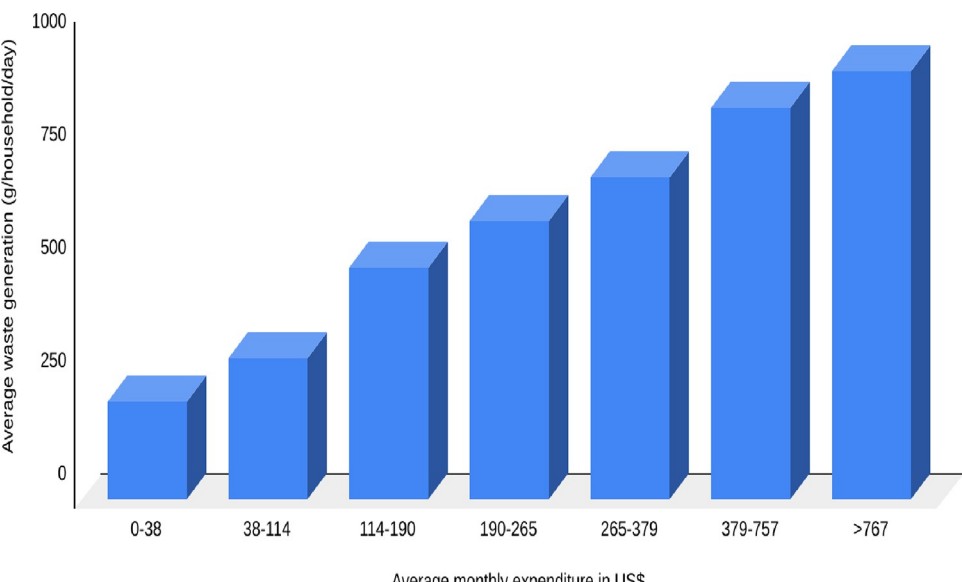

**Fig 3. Waste generation by household with average monthly expenditure.**

pharmaceuticals, 0.4% electronics, and 12% other wastes (Fig 4). The highest fraction of paper and paper products and plastics came from schools; plastics from shops and hospitals; pharmaceuticals from medical shops and hospitals and organic waste from hotels and restaurants. Compared to household and institutional waste, a high rate of glass in commercial waste was recorded. Institutional wastes included 45% paper and paper products, 22% organic waste, and 21% plastics. This study also found that 43% of commercial wastes were organic wastes, 23% were paper and paper products, and 22% were plastics. Hospital waste, obsolete pesticides, and a few industrial wastes were the most common sources of hazardous waste in BM. It was reported that the waste generated by one of the biggest hospitals in Lamjung in BM was 141.765 kg/day, consisting of 91.94% nonhazardous waste and 8.060% hazardous waste [3]. In BM, most hospitals, health posts, and health clinics did not have incinerators. These wastes were not collected and disposed of separately, they were mixed with MSW. On average, MSW in 53 municipalities consisted of 56% organic waste, 16% plastic waste, and 16% paper and paper products [9].

## Total quantity of waste generated in BM

Waste generation in a municipality is influenced by various factors. Among them, the size and density of the population are crucial determinants, as larger urban areas typically produce more waste [11]. Economic development is another significant factor, as it leads to increased production of goods and services, consequently resulting in higher waste generation. Moreover, consumer behavior and lifestyle choices have a substantial impact on the types and quantities of waste produced. Societies that heavily rely on packaged goods and single-use products tend to generate more waste compared to those practicing sustainable consumption [1, 9, 11]. These factors collectively shape the waste generation patterns in a given municipality.

In Besisahar municipality, solid waste generation was determined by considering several factors. The average per capita household waste generation was determined by multiplying the average waste generation by the total number of residents, resulting in an estimated 7427.75 kg/household/day. This household waste constituted approximately 46.02% of the total waste

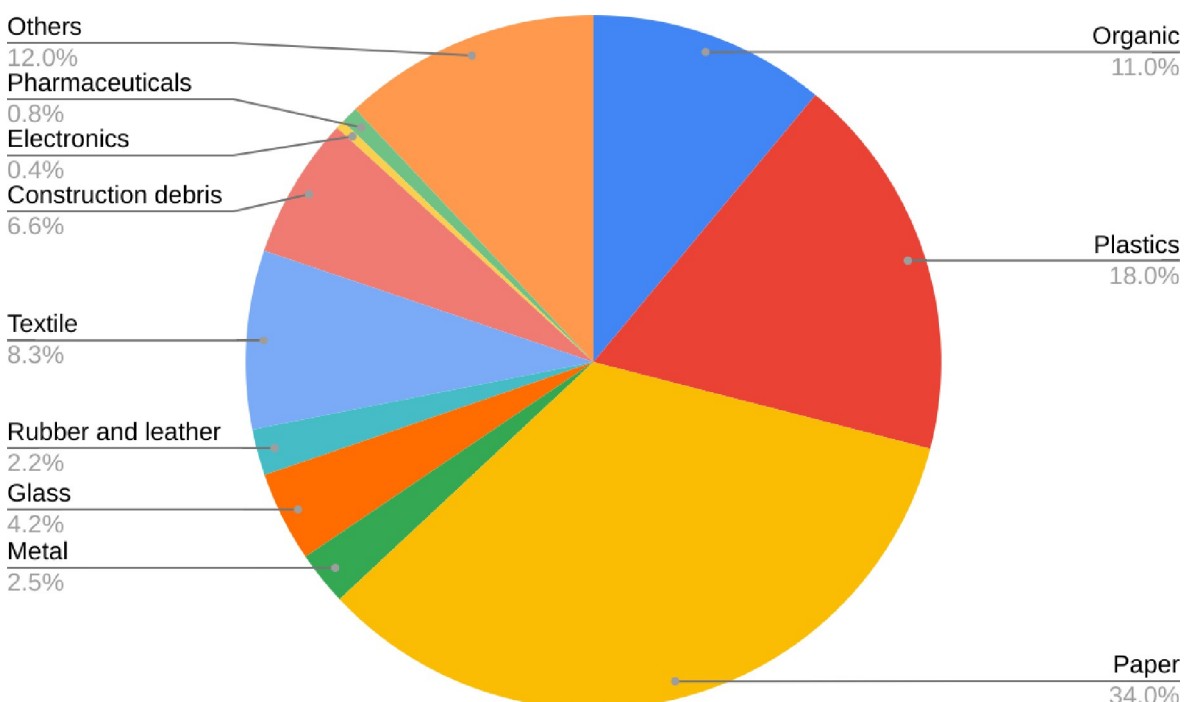

**Fig 4. Composition of institutional, commercial, and industrial wastes in BM.**

generated in the municipality. Apart from households, other sectors also contributed to waste generation such as hospitals generated 124.96 kg/day (accounting for 0.77% of the total waste), hotels and restaurants contributed 3,270 kg/day (20.28%), commercial shops accounted for 5,000 kg/day (30.98%), academic institutions generated 116.9 kg/day (0.72%), banks and financial institutions produced 34.4 kg/day (0.21%), government and private offices generated 64 kg/day (0.39%), and industries contributed 100 kg/day (0.61%). Collectively, all these sectors contributed approximately 16138.01 kg/day in the entire BM.

In Bhimeshwor Municipality, the total daily waste generation was estimated to be around 15,011 kg/day, which aligns with the findings of the waste analysis conducted in this municipality. Likewise, Janakpurdham Sub-Metropolitan City generated a total waste of 78,273 kg/day. In Pokhara Metropolitan City, the total daily waste generation was 134,500 kg/day. This was composed of 94,150 kg/day (70%) from households, 6,725 kg/day (5%) from businesses, 6,725 kg/day (5%) from industries, 13,450 kg/day (10%) from educational institutions, 4,559.6 kg/day (3%) from health institutions, and 8,890.4 kg/day (7%) from other sectors. On an annual basis, the average total waste collected per municipality in the years 2073/74, 2074/75, and 2075/76 was approximately 2,231 metric tons (mt), 2,164 mt, and 2,233 mt, respectively, according to data from the Government of Nepal. This translates to an average daily waste collection per municipality of 6.1 mt, 5.9 mt, and 6.1 mt, respectively, over the course of those three years [20].

Our findings indicate that waste generation per capita per day is higher in wards located in city areas, attributed to factors such as higher population density, increased economic activities, and different consumer behaviors prevalent in urban areas. Indeed, the effectiveness of waste management systems and practices implemented by a municipality can significantly influence the total quantity of waste generated [1]. To address this issue, implementing efficient waste management strategies, such as recycling, composting, and waste reduction

initiatives can play a significant role in minimizing waste generation [11]. Additionally, the regulatory framework and policies established by local governments are vital in promoting sustainable waste management practices. Legislation mandating waste segregation, promoting recycling, and imposing waste disposal fees can serve as incentives for residents and businesses to adopt more environmentally friendly practices, leading to a reduction in waste generation [9]. By considering these factors and implementing appropriate measures, municipalities can work towards improving waste management and achieving a reduction in overall waste generation, contributing to a more sustainable and cleaner environment.

### Practices of SWM in urban areas

The municipality and private waste collectors are responsible for the collection of waste from urban areas of BM. The private waste collectors collect solid waste from the streets, collection centers, and dustbins in business and residential areas. After collecting the waste, it is transferred to a dumping site for disposal. Solid waste collection and transportation systems in urban areas of BM are presented in Fig 5. WM employees are responsible for collecting and transporting solid waste from urban areas, where the roads are accessible by the schedule of the area. Thus, the waste is gathered from different collection points and finally transported to the dumping sites on the bank of the Marshyangdi River. However, waste generators are charged a waste collection fee. Even though there are rules and regulations governing the management of solid waste from the point of generation to disposal, some households, and commercial and industrial solid waste generators were found to burn and bury their waste using

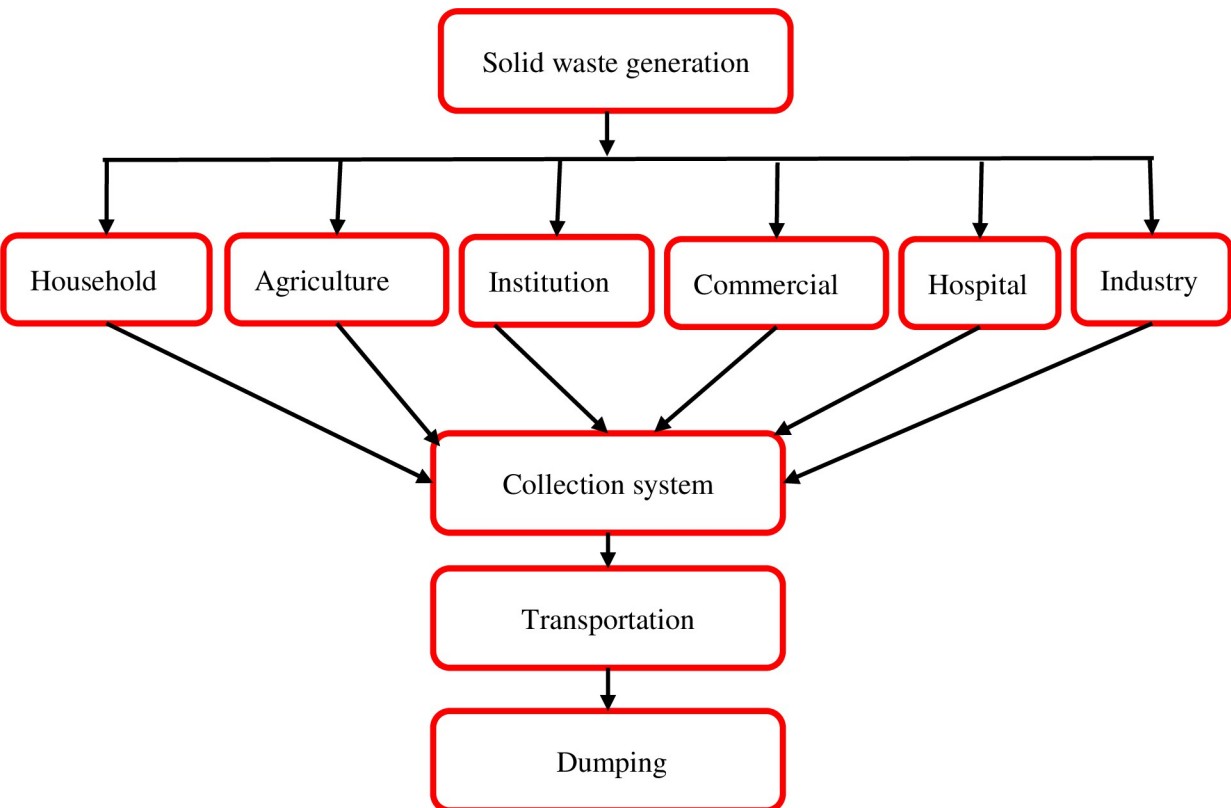

**Fig 5. Flow chart of typical solid waste collection and transportation systems in urban areas.**

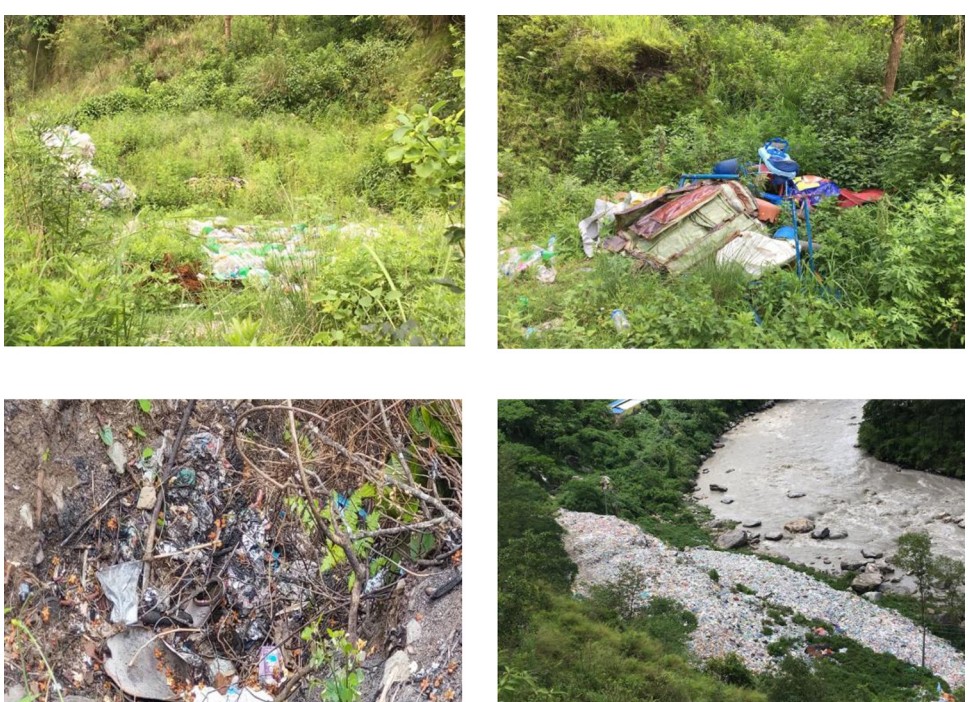

**Fig 6. Disposal and dumping of solid wastes in various locations around BM.**

unfriendly methods to control environmental pollution. MSW is not currently subject to recycling or recovery practices in BM.

In Fig 6, it is evident that waste is being dumped in open areas without any engineering or scientific methodology. Almost all raw wastes from BM are dumped on the riverbanks, whereas villagers dump their household wastes near public places, spring banks or their land or burn. In Besisahar City a tractor with a 2.5-m$^3$ container is employed in the transportation of solid waste for 25 tips per week by the private waste collector, authenticated by the municipality. There was an estimated 4,285 kg/day of solid waste generated in urban areas of BM. Interestingly, 26.55% of solid waste was generated in urban areas compared to semi-urban and village areas. In our study, it was not observed that the minimization, reuse, and recycling of municipal wastes. The results of this investigation demonstrated the municipal system for managing solid waste to be wholly ineffective. To stop the BM from getting worse, immediate action must be taken to improvise. The municipality has also concurred with the statement and is aware that the existing practice is unhealthy and harmful to the environment. It can be suggested that the best method of disposing of solid waste is to compost it and use it as a soil amendment. As a result, less waste will need to be hauled to the landfill, prolonging its life [11]. When a large fraction of solid waste is organic, composting is one of the most sustainable methods of managing it [1]. A study by Timilsina [12] found that 21% of the waste from Kathmandu could be recycled and 62% of it could be composted. As an alternative to landfilling, aerobic composting is becoming increasingly important around the world [1] and is suggested in the BM.

In the Kathmandu Valley, it has been estimated that approximately 3% (20 tons/day) of the municipal solid waste is disposed of through open burning. This releases harmful particulate matter, gaseous pollutants, and toxic substances into the atmosphere, contributing to air pollution and environmental degradation [26]. Similarly, in Bhimeshwor Municipality,

approximately 53% (7,951 kg/day) of the waste generated is collected and disposed of in landfills [20]. Proper landfill management is important to prevent environmental contamination and potential health hazards associated with improper waste disposal. These figures underscore the significance of adopting effective waste management practices, such as waste reduction, recycling, and sustainable disposal methods.

The combination of biodegradable and non-biodegradable wastes leads to the production of methane gas during anaerobic digestion and the release of harmful substances from waste dumps, which can contaminate groundwater resources. To address this issue, specific recyclable materials such as plastic containers and packaging, cans, glass, and PET bottles, used paper, clothes, and oversized waste items need to be collected separately for reuse or recycling purposes, while non-recyclable waste, on the other hand, should undergo incineration. Unfortunately, many landfill sites in Nepal fail to meet scientific or health standards, and waste is not treated prior to disposal, resulting in the creation of hazardous dumpsites [9].

### Response to the current waste disposal system

Solid waste composition plays an important role in deciding which disposal method to adopt for a particular community from a technical perspective. Waste disposal activities in the municipality at present are not technically feasible or environmentally friendly. Table 10 shows statistical parameters for response regarding waste disposal in BM. The majority of the respondents (60%) disagreed with the view that the current dumping sites along Marsyangdi and other small riverbanks, springs, and public areas should not be used for waste disposal. It was agreed by 48% of respondents that waste disposal along riversides impacts heritage, culture, religion, and the environment along with human health. Similarly, 50% of respondents agreed that waste should be separated before being disposed of. Only 15% of the respondents had no idea about waste treatment before disposal. There were approximately equal percentages of respondents who agreed or disagreed with the performance of private waste collectors. However, 33% of respondents agreed that BM should play a key role in WM practices. Many respondents to the questionnaire survey had no knowledge of sustainable WM systems. Based on the results of this study, the municipality does not possess the necessary policies, finances, and technical capacities to establish and maintain a new landfill site.

**Table 10. Statistical parameters for response regarding waste disposal in BM.**

| Questionnaire | No. of participants (n = 230) | | | | | | | | | | Median | AVEDEV | STDEV |
|---|---|---|---|---|---|---|---|---|---|---|---|---|---|
| | Strongly agree | % | Agree | % | Strongly disagree | % | Disagree | % | No knowledge | % | | | |
| Suitable for dumping areas on the riverbank | 6 | 2.60 | 18 | 7.82 | 57 | 24.78 | 138 | 60 | 11 | 4.78 | 18 | 41.2 | 55.21 |
| Impact of dumping on river banks | 48 | 20.86 | 110 | 47.82 | 22 | 9.56 | 34 | 14.78 | 16 | 6.95 | 34 | 26.4 | 37.8 |
| Segregation of waste before dumping | 58 | 25.21 | 115 | 50 | 14 | 6.08 | 18 | 7.82 | 25 | 10.86 | 25 | 32.4 | 42.29 |
| Waste treatment before dumping | 35 | 15.21 | 92 | 40 | 23 | 10 | 46 | 20 | 34 | 14.78 | 35 | 18.4 | 26.97 |
| Private waste collector activities on dumping | 19 | 8.26 | 94 | 40.86 | 34 | 14.78 | 69 | 30 | 14 | 6.08 | 34 | 28.4 | 34.38 |
| Municipality in waste disposal activities | 28 | 12.17 | 76 | 33.04 | 46 | 20 | 69 | 30 | 11 | 4.78 | 46 | 21.2 | 27.28 |
| Implementation of SDGs | 2 | 0.86 | 7 | 3.04 | 46 | 20 | 83 | 36.08 | 92 | 40 | 46 | 33.2 | 41.65 |

## Impact on environment and public health

Waste treatment and final disposal systems in BM are primarily open dumping and burning, and the management of solid waste is made worse by unsustainable practices that increase environmental degradation and the spread of diseases. There is a direct correlation between the composition of waste and the methods of disposal [11]. As a result of the combustion and decay of solid wastes in open spaces, gaseous emissions, particulates, and volatiles are released into the atmosphere, and seepage leads to biological and chemical contamination of soil and groundwater [14, 30, 31]. However, the methane generated during the combustion of organic waste contributes to an increase in global warming and climate change [32]. Furthermore, high organic contents in solid waste will promote the growth of microbial pathogens and cause infections and diseases in surrounding people as well [33]. Medical waste from hospitals, clinics, and other sources can be highly hazardous since it may contain infectious diseases or toxic chemicals. Landfill leachate from open dump sites contains high concentrations of fluoride, chloride, nitrogen, ammonium, toxic metals, organic carbons, biological oxygen demand (BOD), and chemical oxygen demand (COD) [34, 35], as well as methane, carbon dioxide, carbon monoxide, nitrogen, hydrogen sulfide, and ammonia, which are some of the landfill gasses produced by traditional landfill practices [1]. Water sources can be contaminated by leachate, resulting in a negative impact on agricultural resources. The burning of solid wastes and some microbes has been shown to cause several health problems such as skin, eye, and nasal irritations, digestive, allergies, and psychological and respiratory problems [36]. Besides that, uncontrolled hazardous and electronic wastes have been linked to a variety of health problems, including chemical poisoning, low birth rates, cancer, nausea, vomiting, malformations, and neurological disorders [37]. It is believed that mixing healthcare products and biomedical waste with domestic waste and disposing of it at dumping sites without treatment increases the risk of Hepatitis B and HIV [31].

Based on the analysis of 20 water samples taken in Besisahar city and its surroundings, Aryal [38] reported that most of the samples exceeded the threshold standards for *Escherichia coli*, total coliforms, iron, manganese, and selenium, possibly due to improper SWM. In another study by Ono et al. [39], water samples collected from the Kathmandu Valley's water supply system contained one or more bacterial species in 43 out of 57 samples, of which 51% contained *E. coli* in 22 out of 43, and they concluded that diarrhea was caused by contaminated drinking water. It is currently common practice to landfill solid waste on the banks of rivers, streams, and open fields without considering their negative impact on water resources or the health of surrounding residents.

## Economic benefit from solid waste

Waste management costs can be reduced by recycling the waste as a resource, thus decreasing the amount of waste that needs to be disposed of. Organic waste composting can also significantly reduce the costs and environmental impacts in households, in the community, and at the municipal level [11]. As well, organic waste separation increases the market value and recycling potential of inorganic waste such as paper and plastics.

The private waste collectors are not a company but got a bid from the municipality. They are obtaining money from every home and other organizations as management fees. There is no facility for precious waste segregation for direct-to-market and utilization processes. It was observed that this is a very weak side of the BM and waste collector. Recycling municipal waste and producing energy from organic waste should be prioritized by the municipality. This study revealed that the municipality will earn money from the recyclable waste and will get energy like cooking gas from the waste, which can make up for the energy consumption.

Several efforts have been made to develop scientific environmental controls and build a progressive landfill in BM. It has been found that the operation of distant disposal sites has been associated with high transportation costs, making it impossible to maintain environmentally friendly waste disposal practices. Additionally, reuse and recycling not only provide livelihoods for those concerned but also significantly reduce the amount of waste at dumping sites. SWM must be sustainable by involving the private sector in WM, educating the public on environmental issues, and enforcing strict environmental laws [40].

## Linkages with SDGs

Open dump sites pose a serious health threat to people living nearby waste collection centers and dumping sites, as well as releasing solid waste into water bodies, causing environmental contamination [41]. Target 3.2 aims to end preventable deaths of children under 5 years of age. The consumption of contaminated water due to the mismanagement of solid wastes can cause chronic and enteric diseases. In Nepal, enteric fever and diseases related to drinking water are the main culprits for the deaths of children under 5 years [42]. Target 3.9 refers to reducing the number of deaths and diseases due to hazardous chemicals, pollution, and contamination of air, water, and soil [13]. Aryal et al. [3] reported that medical waste disposal in BM is often problematic. It is important to properly manage medical waste within health facilities, such as by incineration, sterilization, or shredding, to reduce the spread of infections and pathogens. A lack of proper hazardous waste separation practices still poses a significant risk of illness and infection to garbage collectors in the study area. The fact that children and household members continue to suffer from chronic diseases indicates that SDG 3 has not been achieved.

Target 6.3 addresses enhancing water quality by reducing pollution, eliminating dumping, minimizing releases of chemicals and hazardous substances, substantially increasing recycling, and ensuring safe reuse [13]. It is recommended by Rodic and Wilson, [43] that all wastes, whether chemical or biological, be managed environmentally soundly, particularly those that end up in rivers, streams, or underground water. For instance, plastics do not degrade but turn into nanoparticles less than 2.5 mm in size, which then enter the food chain. The consumption of drinking water that is contaminated with nano-plastics contributes to different cancers and hormonal disorders [44]. The majority of solid waste has been managed near water resources in our study area, making it almost impossible to achieve this goal in the near future [45]. This study suggests that MSWs from BM contain a high level of organic waste, which will be turned into organic fertilizers as well as energy to achieve Target 6.3.

Effective household SWM practices play a vital role in aligning with SDG 11, "sustainable cities and communities". These practices contribute significantly to environmental protection, particularly at the local and municipal levels. However, several regions in Asia are currently facing challenges in meeting Target 11.6, which focuses on improving waste management in urban areas [46]. Since SWM policies and services are typically under the responsibility of local governments, enhanced support at the municipal level can exert pressure on higher levels of government, compelling them to take action and allocate additional funding. This proactive approach directly addresses one of the major barriers to progress in waste management.

Target 13.3 refers to increased knowledge and capacity to deal with climate change [13]. MSW disposal facilities in developing countries are often open dumpsites, and the organic waste releases greenhouse gases, mainly methane, which contribute contributes to air pollution and climate change. Rodic and Wilson, [43] mentioned that the disposal of solid waste in a well-controlled manner and the reduction of open burning can reduce climate change. A solution that can be suggested is to reduce waste to dumping sites and burn it as well as recycle corresponding wastes to achieve SDG 13, target 13.3.

Hence, local policymakers must acknowledge the interdependence of SWM targets and other SDGs. Recognizing that advancements in WM can drive progress in other SDGs, and vice versa, is essential. This understanding enables local policymakers to advocate for comprehensive WM strategies that not only benefit the environment but also contribute to broader sustainable development goals. However, to achieve these objectives, significant investments are necessary. Adequate funding is required to address the lack of robust SWM systems and to support educational campaigns emphasizing the importance of waste separation. Without sufficient financial commitments, inadequate waste management practices will persist, impeding sustainable development efforts, especially in rapidly expanding urban areas across Asia.

## Conclusions and future recommendations

The management of solid waste is still a major challenge for both local governments and private waste managers in developing economies. As a result of this work, it provides a clearer picture of WM by primarily collecting and separating waste at household and local levels. As determined by the waste composition study, household solid waste was composed of 68.03% organic, 8.13% paper, 5.5% agricultural, 5.21% plastic, 3.81% construction debris, 2.72% textile, 1.01% glass, 0.54% metal, and 0.1% rubber, 0.1% pharmaceutical, 0.05% electronic, and 4.8% other materials. A study conducted in BM revealed that household solid waste was generated at 197.604 g/capita/day. The waste composition in institutional and commercial establishments revealed 11.02% organic waste, 34% paper and paper products, 18% plastics, 8.32% textiles, 6.6% construction debris, 0.8% pharmaceuticals, 0.4% electronics, and 12% other wastes. The results obtained from Pearson correlation coefficients indicated robust correlations among the parameters of different types of wastes generated in BM. Around 4.285 tons-solid waste/day were generated in urban areas, while 16.138. tons-solid waste/day were estimated for the whole municipality. Most of the respondents (60%) disagreed with the view that the current dumping sites along Marsyangdi and other small riverbanks, springs, and public areas should not be used for waste disposal A sustainable resource recovery process can be achieved by optimizing waste separation and reducing the need for waste sorting and cleanup. A better WM system will result in higher living standards and better health for many people with limited services, prevent hazardous materials from entering water resources, contribute significantly to the reduction of climate change, and help restore the environment. A WM regulation should be implemented by BM and its authorities, and a community awareness program would make the community responsible for managing the waste as well. As a result of the study, the following recommendations are made for improving the current SWM system in BM.

1. Government agencies, institutions, and non-governmental organizations need to continuously educate the community about the effective handling of solid waste at the household level.

2. A change in the moral attitude of people should lead to the separation and segregation of waste at the location.

3. A sustainable approach to managing solid waste in BM is to compost organic waste and use it as organic fertilizer.

4. Increasing public awareness about recycling materials such as metals and plastics at the point of origin.

5. Private companies must be involved in SWM to achieve sustainability.

6. Municipalities should implement strong rules and regulations to prevent the illegal dumping of solid waste along riverbanks and in public places.

## Supporting information

**S1 File. Survey questionnaire (English version).**
(DOCX)

## Acknowledgments

The authors are very grateful for the support from the BM, private waste collectors, and residents for the information they provided through interviews.

## Author Contributions

**Data curation:** Mahendra Aryal, Sanju Adhikary.

**Formal analysis:** Mahendra Aryal, Sanju Adhikary.

**Investigation:** Mahendra Aryal, Sanju Adhikary.

**Methodology:** Mahendra Aryal, Sanju Adhikary.

**Resources:** Mahendra Aryal, Sanju Adhikary.

**Supervision:** Mahendra Aryal.

**Validation:** Mahendra Aryal, Sanju Adhikary.

**Writing – original draft:** Mahendra Aryal.

**Writing – review & editing:** Mahendra Aryal.

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
