## [Decision Letter · Decision Letter 0]

31 May 2023

PONE-D-23-04289Solid waste management practices and challenges in Besisahar municipality, NepalPLOS ONE

Dear Dr. Aryal,

Thank you for submitting your manuscript to PLOS ONE. After careful consideration, we feel that it has merit but does not fully meet PLOS ONE’s publication criteria as it currently stands. Therefore, we invite you to submit a revised version of the manuscript that addresses the points raised during the review process. Please submit your revised manuscript by Jul 15 2023 11:59PM. If you will need more time than this to complete your revisions, please reply to this message or contact the journal office at plosone@plos.org. Please include the following items when submitting your revised manuscript:A rebuttal letter that responds to each point raised by the academic editor and reviewer(s). You should upload this letter as a separate file labeled 'Response to Reviewers'.A marked-up copy of your manuscript that highlights changes made to the original version. You should upload this as a separate file labeled 'Revised Manuscript with Track Changes'.An unmarked version of your revised paper without tracked changes. You should upload this as a separate file labeled 'Manuscript'.

We look forward to receiving your revised manuscript.

Kind regards,

Noé Aguilar-Rivera

Academic Editor

PLOS ONE

Journal Requirements:

2. You indicated that ethical approval was not necessary for your study. We understand that the framework for ethical oversight requirements for studies of this type may differ depending on the setting and we would appreciate some further clarification regarding your research. Could you please provide further details on why your study is exempt from the need for approval and confirmation from your institutional review board or research ethics committee (e.g., in the form of a letter or email correspondence) that ethics review was not necessary for this study? Please include a copy of the correspondence as an ""Other"" file.

4. We note that Figure 1a and 1b in your submission contain map/satellite images which may be copyrighted. All PLOS content is published under the Creative Commons Attribution License (CC BY 4.0), which means that the manuscript, images, and Supporting Information files will be freely available online, and any third party is permitted to access, download, copy, distribute, and use these materials in any way, even commercially, with proper attribution. For these reasons, we cannot publish previously copyrighted maps or satellite images created using proprietary data, such as Google software (Google Maps, Street View, and Earth). For more information, see our copyright guidelines: http://journals.plos.org/plosone/s/licenses-and-copyright.

(1) You may seek permission from the original copyright holder of Figure 1a and 1b to publish the content specifically under the CC BY 4.0 license. 

**Additional Editor Comments:**

**(Reviewer 1)**

The research is of great importance and contributes to waste management in Nepal, and works as a reference for policymakers in countries or areas that share similarities in municipal solid waste generation or management. However, the way how the authors display their research is limited to descriptive information, which lacks critical analysis, and key findings are not clearly presented.

Specific comments

Please note that all tables are missing in the manuscript.

It would be helpful if line numbers could be provided.

The last paragraph of the introduction should state the purpose of this paper concisely.

Page 4, BMP: please spell out when it first appears in the text.

Page 5, three decimal places are unnecessary for percentages in the text, and it is better to keep one decimal for percentage data.

Page 5

Please specify the qualitative and quantitative methods in this study.

Page 6

Please specify sources of data when you mention them in the text, for example,

The results of the questionnaire study revealed that 72% of household respondents knew about waste separation, but they did not apply it practically, whereas only 28 % of respondents indicated that they were utilizing such knowledge to separate waste properly at the household level.

It was reported that solid waste collection was implemented in wards numbers 1, 2, 3, 7, 8 and 9.

**Reviewer 2**

The following are the specific comments which need to be addressed before its publication;

It’s very difficult to review the submitted manuscript without adding the line numbers, thus, I commented directly in the attached PDF through annotation tools. Authors must add the line number before submitting the revised version of the manuscript, if any.

What are the differences between this study and others in the literature? The originality/novelty of the paper should be clearly stated in the introduction section.

Manuscript is completely lacking with citation of 2023. In addition, the authors have cited 2-3 references only for the year, 2022. The manuscript should cover the recent literature related to this subject. Introduction is completely lacking the citation from the years 2022; 2023.

Old references may be replaced with recent references.

The discussion needs improvement and should be linked to the findings of the previous reports on this topic. discussion is elaborative but it needs more adequate discussion with supporting latest references. Discussion should be according to the results.

Lack of proper statistical analysis. Authors should perform some advanced statistical analysis for figures 3, 4, and 9. A description of statistical techniques and tests applied to the analyses of data sets must be mentioned.

The visibility of all supplied figures (Fig. 6, 7, and 9) is not good. Authors are advised to provide the figures in high resolution with good font visibility. The size of the fonts is very small, please increase it also.

Reviewers' comments:

Reviewer's Responses to Questions

**Comments to the Author**

1. Is the manuscript technically sound, and do the data support the conclusions?

Reviewer #1: Partly

Reviewer #2: Yes

2. Has the statistical analysis been performed appropriately and rigorously? 

Reviewer #1: No

Reviewer #2: No

3. Have the authors made all data underlying the findings in their manuscript fully available?

Reviewer #1: No

Reviewer #2: No

4. Is the manuscript presented in an intelligible fashion and written in standard English?

Reviewer #1: Yes

Reviewer #2: Yes

5. Review Comments to the Author

Reviewer #1: The research is of great importance and contributes to waste management in Nepal, and works as a reference for policymakers in countries or areas that share similarities in municipal solid waste generation or management. However, the way how the authors display their research is limited to descriptive information, which lacks critical analysis, and key findings are not clearly presented.

Specific comments

Please note that all tables are missing in the manuscript.

It would be helpful if line numbers could be provided.

The last paragraph of the introduction should state the purpose of this paper concisely.

Page 4, BMP: please spell out when it first appears in the text.

Page 5, three decimal places are unnecessary for percentages in the text, and it is better to keep one decimal for percentage data.

Page 5

Please specify the qualitative and quantitative methods in this study.

Page 6

Please specify sources of data when you mention them in the text, for example,

The results of the questionnaire study revealed that 72% of household respondents knew about waste separation, but they did not apply it practically, whereas only 28 % of respondents indicated that they were utilizing such knowledge to separate waste properly at the household level.

It was reported that solid waste collection was implemented in wards numbers 1, 2, 3, 7, 8 and 9.

Reviewer #2: The following are the specific comments which need to be addressed before its publication;

It’s very difficult to review the submitted manuscript without adding the line numbers, thus, I commented directly in the attached PDF through annotation tools. Authors must add the line number before submitting the revised version of the manuscript, if any.

What are the differences between this study and others in the literature? The originality/novelty of the paper should be clearly stated in the introduction section.

Manuscript is completely lacking with citation of 2023. In addition, the authors have cited 2-3 references only for the year, 2022. The manuscript should cover the recent literature related to this subject. Introduction is completely lacking the citation from the years 2022; 2023.

Old references may be replaced with recent references.

The discussion needs improvement and should be linked to the findings of the previous reports on this topic. discussion is elaborative but it needs more adequate discussion with supporting latest references. Discussion should be according to the results.

Lack of proper statistical analysis. Authors should perform some advanced statistical analysis for figures 3, 4, and 9. A description of statistical techniques and tests applied to the analyses of data sets must be mentioned.

The visibility of all supplied figures (Fig. 6, 7, and 9) is not good. Authors are advised to provide the figures in high resolution with good font visibility. The size of the fonts is very small, please increase it also.

6. PLOS authors have the option to publish the peer review history of their article (what does this mean?). If published, this will include your full peer review and any attached files.

Reviewer #1: No

Reviewer #2: No

---

## [Author Response · Author response to Decision Letter 0]

12 Aug 2023

Dear Reviewers,

We would like to express our sincere gratitude for taking the time to review our manuscript. Your thoughtful comments and valuable feedback have been instrumental in enhancing the quality of our work. We have carefully considered each of your suggestions and made the necessary revisions to address the concerns raised. Please, find the detailed responses to your comments in the attached "Answer to Rewiewwers Comments" and "revised manuscript".

Yours

Mahendra Aryal

---

## [Decision Letter · Decision Letter 1]

28 Sep 2023

Solid waste management practices and challenges in Besisahar municipality, Nepal

PONE-D-23-04289R1

Dear Dr. Mahendra Aryal

We’re pleased to inform you that your manuscript has been judged scientifically suitable for publication and will be formally accepted for publication once it meets all outstanding technical requirements.

Kind regards,

Noé Aguilar-Rivera

Academic Editor

PLOS ONE

---

## [Editor Report · Acceptance letter]

11 Mar 2024

PONE-D-23-04289R1 

PLOS ONE

Dear Dr. Aryal, 

I'm pleased to inform you that your manuscript has been deemed suitable for publication in PLOS ONE. Congratulations! Your manuscript is now being handed over to our production team.

Kind regards, 

on behalf of

Dr. Noé Aguilar-Rivera 

Academic Editor

PLOS ONE